# Spectroscopic and Spectroelectrochemical Studies of Hexapentyloxytriphenylene—A Model Discotic Molecule

**DOI:** 10.3390/ijms24086924

**Published:** 2023-04-08

**Authors:** Piotr Ślęczkowski

**Affiliations:** International Centre for Research on Innovative Biobased Materials (ICRI-BioM)—International Research Agenda, Lodz University of Technology, Zeromskiego 116, 90-924 Lodz, Poland; piotr.sleczkowski@p.lodz.pl

**Keywords:** discotic liquid crystal, hexaalkoxytriphenylenes, UV-Vis absorption spectroscopy, cyclic voltammetry, EPR spectroelectrochemistry, molecular aggregation

## Abstract

The electrochemical and spectroelectrochemical properties of the discotic mesogen 2,3,6,7,10,11-pentyloxytriphenylene (H5T) were studied with the use of cyclic voltammetry combined with UV-Vis and electron paramagnetic resonance (EPR) spectroscopy in solution. The UV-Vis absorption spectroscopy of H5T in dichloromethane showed its monomeric state in a concentration range up to 10^−3^ mol dm^−3^. The reversible process of the electrochemical formation of the radical cation was evidenced within the experimentally accessible potential window. The in situ UV-Vis spectroelectrochemical measurements further enabled identification of the product of the redox process and evaluation of the effect of aggregation in the concentration range of 5 × 10^−3^ mol dm^−3^. The results are discussed in the frame of solvent effects on the self-assembly propensity of solute molecules, in a wide range of concentrations. In particular, the crucial role of the solvent polarity is indicated, which contributes to the understanding of solution effects and pre-programming of supramolecular organic materials, in particular anisotropic disc-shaped hexa-substituted triphenylenes.

## 1. Introduction

Since their discovery by Chandrasekhar in 1977 [1], discotic liquid crystals (DLCs) have attracted great attention [2]. One of the main reasons for this is their substantial potential for use as electroactive layers in organic electronic devices [3,4]. The origin of the semiconducting properties of DLCs lies in their ability to form columnar (Col) mesophases (Figure 1a). The shape anisotropy of discotic molecules, consisting of a rigid polyaromatic core substituted with flexible alkyl tails, results in strong *π*–*π* interactions between the aromatic cores, which promotes the cofacial alignment of molecules with a typical intermolecular distance of 3.5–4.5 Å. The intermolecular *π*–*π* stacking gives birth to supramolecular columns, which form a 2D lattice with long-range order and an intercolumnar distance controlled by the length of peripheral alkyl tails, typically in the range of 20–40 Å [5]. Due to the self-healing property, the size of DLCs’ monodomain can reach tens to hundreds of microns, resulting in undisturbed charge transport along the defect-free one-dimensional (1D) molecular wires, which in turn offers a unique platform for hybrid organic-inorganic nanomaterials [6].

One of the most extensively studied DLC central motifs is triphenylene [7], which was reported to support the formation of columnar mesophases for as few as three peripheral substituents [8,9]. Synthetically the most easily accessible substituted triphenylenes are symmetrical 2,3,6,7,10,11-hexaalkoxy derivatives [7], H*n*Ts, where *n* typically stands for the length of the –OC*_n_*H_2*n*+1_ side chain. Functional groups, such as azobenzene [10] or other conjugated moieties [11,12], can be introduced for tailoring of the surface or bulk properties of triphenylene-based materials. Besides the extensive studies on the photophysical [13] and charge transport properties [14] of H*n*Ts, which are crucial for their implementation in organic light-emitting diodes (OLEDs) [15], organic field-effect transistors (OFETs) [16], or organic photovoltaics (OPVs) [17], several aspects connected with molecular order and thin film engineering have been investigated. Due to the liquid-like properties of DLCs, a large amount of studies are concentrated on the self-assembly of H*n*Ts in thin films [18], under geometrical confinement [19], or the growth of nanostructures of different dimensionality [20]. Moreover, to understand the processes occuring at the organic/metal interface, detailed studies describing the formation of self-assembled molecular networks (i.e., physisorbed monolayers) of H*n*Ts on conductive substrates have been reported [21]. Besides understanding thermodynamic as well as kinetic aspects of the interfacial phenomena [22], currently we observe the development of several strategies aimed at improving electronic device performance, e.g., stimuli-directed alignment of DLCs [23].

Since the fabrication of organic active layers for optoelectronic devices preferably involves solution processing, e.g., spin coating [16,17,18], knowledge about the behavior of molecules in solution is highly desirable. The aggregation has a great influence on the electronic properties of solutions of polycyclic aromatic molecules, since functionalities often arise due to electronic coupling between *π*-conjugated systems [24]. The anisotropy of the molecular shape of DLCs not only results in the formation of columnar mesophases, but was evidenced to cause self-association of discotic molecules in solutions [25], as illustrated in Figure 1b. For example, small-angle neutron scattering (SANS) experiments of H*n*Ts with various side-chain lengths (*n* = 5, 9, and 11) in *n*-hexadecane revealed the strong propensity of the studied molecules to self-assemble in solution. In particular, the H5T analog was reported to form rod-like aggregates at concentrations above 10^−3^ mol dm^−3^ [26]. Besides scattering methods, UV-Vis absorption and photoluminescence spectroscopy remain the first choice for studies of aggregtation of H*n*Ts. Gallivan and Schuster reported on an ideal Beer–Lambert law behavior of H6T in *n*-dodecane for concentrations below 10^−4^ mol dm^−3^, with signs of aggregation around 10^−2^ mol dm^−3^ [27]. In the case of H6T in a more polar environment, i.e., dichloromethane, the Beer–Lambert law was also obeyed for concentrations up to 3 × 10^−4^ mol dm^−3^ [28]. Interestingly, studies in cyclohexane revealed a deviation from the Beer–Lambert law for concentrations as low as 10^−5^ mol dm^−3^ [29], most probably due to the strong change in solvent polarity in comparison with dichloromethane [30]. Boden et al. reported that H6T/ethanol and H6T/dichloromethane solutions showed essentially identical absorption spectra with a maximum at 277 nm, strictly obeying the Beer–Lambert law in the concentration range from 10^−7^ to 8 × 10^−4^ mol dm^−3^ [31]. This suggested that, in agreement with previous studies [28,32], H6T remained in the monomeric form. UV-Vis absorption spectroscopy studies of H5T, the closest in the H*n*T homolog series, reported that it is almost identical to H6T. No significant spectral differences were found in *n*-heptane, toluene, 1,2-dichloroethane, and dichloromethane solutions for concentrations up to 2 × 10^−3^ mol dm^−3^ [33]. This was also confirmed by combined experimental and quantum chemical calculation studies [34].

Equally important to the self-assembly and molecular aggregation of DLCs are their electronic properties, such as redox potentials and frontier orbital energies, since they determine charge carrier transport across the interfaces with electrode materials [35] and between different layers in electronic devices [36]. Bushby et al. reported the redox potentials of a variety of functionalized triphenylene-based discogens, determined by cyclic voltammetry and semi-empirical molecular orbital calculations [37]. Radical cations of H*n*Ts were studied in the frame of the potential use of DLC charge transfer (CT) complexes in photovoltaic applications [38]. The prototypical CT discotic complex is formed when electron-rich H5T [13,33,34] or H6T [38] is doped with an electron acceptor e.g., 2,4,7-trinitro-9-fluorenone. The chemical doping of H*n*Ts results in the appearance of an additional CT band in the absorption spectrum, which can be alternatively used for probing of aggregation of molecules. Although previous work has evidenced that the cationic species produced by doping of H6T with AlCl_3_ consist of a single radical cation [31], the spectroelectrochemical properties of H*n*Ts have not been investigated, and detailed studies on electrochemically formed radical cations have not been reported.

To address this issue, the presented paper focuses on the electrochemical and spectroelectrochemical properties of 2,3,6,7,10,11-hexapentyloxytriphenylene (H5T, Figure 1c), a model triphenylene-based DLC. The study presents spectroscopic and spectroelectrochemical characterization using cyclic voltammetry measurements combined with UV-Vis absorption and EPR spectroscopy in solution, for various H5T concentrations. The concentration limit for H5T aggregation in dichloromethane is evidenced from the in situ spectroelectrochemical measurements. The results are discussed in the frame of solvent effects on the self-assembly propensity of solute molecules, in a wide range of concentrations. In particular, the crucial role of the solvent polarity is indicated.

## 2. Results and Discussion

The absorption spectra of dilute solutions of H*n*Ts were previously reported as very structured, which most commonly originates from various electronic transitions and/or vibrational progressions occuring in the system [34]. Figure 2a shows UV-Vis absorption spectra of solutions of H5T in dichloromethane, revealing their unaltered features in the concentration range between 10^−7^ and 10^−5^ mol dm^−3^. The spectra show a strong absorption maximum at 279 nm, corresponding to an S_0_ → S_4_ transition [34], with a molar absorption coefficient (ε) around 1 × 10^5^ dm^3^ mol^−1^ cm^−1^. The absorbance at 279 nm strictly obeys the Beer–Lambert law in the concentration range between 1 × 10^−7^ and 5 × 10^−5^ mol dm^−3^ (Figure 2b), suggesting that H5T is present in the monomeric form over this concentration range. Only slight deviation from ideal Beer–Lambert law behavior is observed for the concentrations around 10^−4^ mol dm^−3^. The monomeric state of H5T is further supported by Figure 2c showing the normalized spectra in the range 10^−7^–10^−5^ mol dm^−3^, which appear identical after superimposing. In addition, the ratio between the two main absorption peaks, at 279 nm and 309 nm (corresponding to an S_0_ → S_3_ transition [34]), exhibits a constant value around 4.8 (Figure 2d). Thus, it can be concluded that H5T does not form aggregates in dichloromethane in the 10^−7^−10^−5^ mol dm^−3^ concentration range, which is in agreement with previous reports on UV-Vis absorption of H5T in dichloromethane [33].

Cyclic voltammetry (CV) was employed to study the redox processes of H5T. The electrochemical potential was calibrated using the ferrocene/ferrocenium (Fc/Fc+) couple by conducting the measurement of ferrocene solution before and after the measurements of the H5T sample solution. The energy level of the Fc/Fc+ couple was assumed to be equal to −4.8 eV with respect to a vacuum [39]. The half-wave potential of the Fc/Fc+ couple (*E*_1/2,Fc/Fc+_ = 0.48 V) was evaluated from the equation *E*_1/2,Fc/Fc+_ = (*E*_pc_ + *E*_pa_)/2, where *E*_pc_ and *E*_pa_ denote the cathodic and anodic potentials, respectively.

Anodic cyclic voltammograms of H5T are given in Figure 3a, revealing one reversible oxidation wave, with a half-wave oxidation potential value of *E*_1/2,ox_ = 1.05 V versus Ag/Ag^+^. The HOMO energy level of H5T was then calculated to be equal to −5.37 eV using the equation *E*_HOMO_ = −(*E*_1/2,ox_ − *E*_1/2,Fc/Fc+_) –4.8 eV. No reversible reduction wave was observed in the range of potentials available in the experiment, indicating the only electron-donor character of the H5T compound. In such a case, the LUMO energy level of H5T was calculated by subtracting the optical bandgap value from the HOMO energy level. The optical bandgap (*E_g_* = 3.7 eV) was estimated from the onset of the absorption as depicted in Figure 3b and yielded a LUMO energy level equal to −1.67 eV. Table 1 presents the values of HOMO and LUMO energy levels and optical HOMO–LUMO gap. It should be emphasized that the presented results are in agreement with previously reported studies of H5T in dimethylformamide [40], despite the fact that the method used here provides an approximate value of the LUMO level [41]. The reversible oxidation process revealed by the cyclic voltammetry studies is interpreted as the H5T radical cation formation. No features of electrochemical polymerization could be found in a 25-cycle scan of the 1 ×10^−3^ mol dm^−3^ H5T in the electrolyte solution, which supports the reversibility of the radical cation formation in the range between 0.00 and 1.20 V (Appendix A). The absence of chemical processes within the timescales of the experiment is also confirmed by the results of the electrode background scans in bare electrolyte solutions (not containing H5T), performed before and after CV of H5T. Since they exhibited no differences, it is assumed that no chemical species were deposited on the electrode in the 0.0–1.2 V potential window (Appendix A). An additional current wave around 1.3 V can be noticed, which is followed by a strong increase in current, when higher potentials are applied (red curve in Figure 3a). Similarly to the 0.0–1.2 V potential window, a multiple scan shows no significant increase in the height of current peaks, excluding the electrochemical polymerization (Appendix A). However, the electrode background scan in bare electrolyte solution performed after CV of H5T shows a broad oxidation peak around 1.5 V that does not appear during the initial scan in bare electrolyte solution (Appendix A). No clear reverse reduction peak implies an irreversible electrochemical step occurring at higher potentials, i.e., following the primary electrochemical redox process at *E*_1/2,ox_ = 1.05 V. In addition, a series of CV experiments was also performed in mixtures of DCM and acetonitrile solvents, which did not show significant changes in H5T oxidation and the related half-wave potential (Appendix A).

The oxidation of H5T was further investigated by performing spectroelectrochemical studies, which consisted of collecting UV-Vis absorption spectra after applying a fixed potential for a time interval of 1 min. In agreement with cyclic voltammetry data, applying potentials up to 1.0 V did not result in the appearance of any spectral changes with respect to the initially collected absorption spectrum, i.e., collected at 0.0 V (Appendix A). However, if potentials higher than 1.0 V were applied, the UV-Vis absorption spectra revealed changes connected with electrochemical generation of radical cations of H5T. As shown in Figure 4a, the maximum at 279 nm exhibits a successive decrease as higher potentials are applied. In addition to an isosbestic point located at 290 nm, a significant increase in absorption at longer wavelengths can be noticed. This is in agreement with the fact that radical cations of aromatic compounds tend to absorb at longer wavelengths than the parent neutral compound [42]. The electrochemical generation of H5T radical cations is fully reversible in the range of potentials up to 1.35 V. This is revealed by UV-Vis spectra, which appear identical when collected at 0.0 V prior to and after applying 1.35 V (represented by black curves in Figure 4b). However, the reversibility of H5T radical cation generation for the concentration of 1 × 10^−3^ mol dm^−3^ does not hold for higher concentrations. For example, red curves in Figure 4b show UV-Vis spectra of 5 × 10^−3^ mol dm^−3^ collected at 0.00 V prior to and after applying 1.35 V. The absorption around the 310 nm maximum and in the 340–440 nm range appears stronger after subjecting the sample solution to potentials of 1.35 V, which is highlighted by blue arrows in Figure 4b. Interestingly, the partial irreversibility of the electrochemical process is, most probably, connected with the aggregation of H5T occuring at higher concentration. This can be inferred from the fact that the ratio between the two main absorption peaks (A^279nm^/A^309nm^) equals 3.0 for the 5 × 10^−3^ mol dm^−3^ solution. This value is substantially smaller than 4.6 found for the 1 × 10^−3^ mol dm^−3^ solution, as well as the lower concentrations presented in Figure 2d. It was previously reported for the discotic molecules that their aggregates can display electronic transition in roughly the same spectral region as the monomeric species, however, with drastically different relative intensities [25].

Similarly to the 1 × 10^−3^ mol dm^−3^ case, the UV-Vis spectra of the 5 × 10^−3^ mol dm^−3^ H5T solution reveal a successive decrease in the absorption maximum at 279 nm as potentials higher than 1.0 V are applied (Figure 4c). An isosbestic point at 290 nm and an increase in absorption at longer wavelengths can also be noticed. However, the higher concentration results in the appearance of quantitative spectral changes, e.g., absorption at 305 nm becoming the global maximum when potentials >1.25 V are applied (Figure 4c). The steeper decline of the absorption peak at 279 nm (A^279nm^) for the more concentrated sample is depicted in Figure 4d, where the normalized A^279nm^ values for 1 × 10^−3^ mol dm^−3^ and 5 × 10^−3^ mol dm^−3^ samples are represented by the black circle and red square data points, respectively. For the potentials up to 1.1 V (α regime, Figure 4d), the A^279nm^ remains almost unaltered for both concentrations. In the 1.15–1.3 V applied potential range (β regime), a significant difference between the values of normalized A^279nm^ can be noticed, which is again reduced for applied potentials larger than 1.3 V (γ regime). This more abrupt change in absorption spectra (and absorption maximum A^279nm^) can be considered a cooperative effect, resulting from the aggregation of H5T molecules in dichloromethane at a concentration of 5 × 10^−3^ mol dm^−3^.

Although the existence of both thermotropic and lyotropic mesomorphism in disc-shaped compounds is rather unusual, some triphenylene-based [43] and also larger discotic molecules [44] have been reported to show lyomesomorphism, i.e., concentration-dependent aggregation. For example, the initially formed small columnar stacks of discotic molecules in *n*-dodecane can behave similarly to molecules in a calamitic nematic phase and align accordingly. In addition, it was experimentally evidenced that in more polar solvents than *n*-dodecane, the aggregation of H5T molecules starts already in solution. In particular, studies using a combination of atomic force microscopy and dynamic light scattering revealed the appearance of molecular aggregates of H5T with sizes ranging from 30 nm to 1.5 µm [45]. Interestingly, the characteristic sizes of the aggregates were found to correlate best with the relative polarity parameter of the solvent used, rather than with the molecular dipole moments or permittivity of the solvents. The relative polarity parameter (as described by Reichardt [46]) for toluene, tetrahydrofurane, and chloroform used by Duzhko et al. is equal to 0.099, 0.204, and 0.254, respectively [45]. The relative polarity parameter of dichloromethane, used in the present study, is even larger and equals 0.309 [46]. Thus, the propensity of H5T to aggregate is likely to be preserved in CH_2_Cl_2_ and in particular in the concentration range between 1 × 10^−3^ and 5 × 10^−3^ mol dm^−3^, which corresponds to the 0.06–0.27 wt% range. This is in agreement with the fact that the aggregation in solvents of lower relative polarity parameter occurs in the range 0.01–0.1 wt% [45].

Finally, the electroactive properties of H5T were studied by in situ EPR spectroelectrochemistry (Figure 5). The combination of EPR spectroscopy with electrochemical methods allows the characterization of charged/radical species generated during the redox process as well as the determination of the related electrochemical mechanisms [47]. Figure 5a shows the sequence of EPR spectra of the 1 × 10^−3^ mol dm^−3^ H5T solution under electrochemical oxidation, which reveals the generation of radical cations. The first signs of non-silent EPR spectra emerge when more than 0.7 V is applied (Figure 5b), while the pronounced EPR signal appears for an applied potential equal to 0.9 V. It exhibits a single EPR line, which suggests that the product of oxidation is paramagnetic, i.e., it has an unpaired electron. Since the conjugated core of H5T contains 18 carbon atoms, the unpaired electron interacts with a large number of nuclei, resulting in its delocalization. Therefore, the individual spectral line overlaps and a broad EPR signal is registered [48]. The peak-to-peak width (ΔB_pp_) and the value of the “g-factor” are equal to 0.380 (±0.01) mT and 1.9969, respectively. Interestingly, the EPR signal intensity was found to increase if potentials between 0.9 V and 1.1 V were applied, but when potentials higher than 1.1 V are applied, EPR signal intensity decreases, as depicted in Figure 5c. This is also depicted in Figure 5d, which shows a plot of the relative spin concentration vs applied potential. In agreement with the EPR spectra, a sharp rise in spin number is observed for the applied potentials between 0.9 and 1.1 V. Subsequently, it is followed by a decrease in spin number for potentials above 1.15 V. This decrease clearly demonstrates that further oxidation results in a decline of spin concentration, due to the formation of dicationic species that are EPR silent. This scenario was previously reported for both low-molecular weight and polymeric *p*-type organic semiconductors [49].

The occurring changes in EPR spectra are accompanied by an increase in absorption in the 300–435 nm range, which is shown in Figure 4. Moreover, H5T’s oxidation to the radical cation results in the appearance of an additional absorption band around 825 nm, which is shown in Figure 6a. The 825 nm absorption band revealed similar behavior to the EPR relative spin concentration. Figure 6b shows an increase and a subsequent decrease in A^825nm^, for applied potentials in the range of 1.0–1.2 V and higher than 1.2 V, respectively. Thus, it was evidenced that the NIR localized 825 nm absorption band is an efficient way of monitoring the H5T oxidation state.

## 3. Materials and Methods

### 3.1. Discotic Liquid Crystal

2,3,6,7,10,11-hexapentyloxytriphenylene (H5T) was synthesized and characterized within the Institut Parisien de Chimie Moleculaire, Sorbonne Université (Paris, France). Additional purification by column chromatography was performed, and material from several batches was used for further studies.

### 3.2. UV-Vis Absorption Measurements

Steady state absorption spectra in UV-Vis were recorded using a Jasco V-630 double beam spectrophotometer (JASCO International Co. Ltd., Tokyo, Japan). The measurements were conducted using a pair of quartz cuvettes of either 0.2 cm or 1 cm optical path, with the corresponding pure dichloromethane (CH_2_Cl_2_) solvent as a reference (POCh, Gliwice, Poland, 99.8% pure HPLC grade, used as received). All measurements were recorded at 25 °C.

### 3.3. Cyclic Voltammetry

Cyclic voltammetry (CV) measurements of H5T were carried out using CH Instruments–660C electrochemical workstation (CH Instruments Inc., Austin, TX, USA) with a conventional three-electrode system. The electrolyte solution consisted of a 0.2 mol dm^−3^
*tert*-butylammonium tetrafluoroborate supporting electrolyte (Bu_4_NBF_4_, Sigma-Aldrich, St. Louis, MO, USA, 98% pure, used as received) in CH_2_Cl_2_ (POCh, Gliwice, Poland, 99.8% pure HPLC grade, used as received). Sweep rate was equal to 0.1 V s^−1^. Prior to the measurements, the solution was bubbled with argon for at least 10 min, to ensure an oxygen-free environment during the recording of the electrochemical data. All measurements were recorded at 25 °C. Platinum electrodes were used both as the working and auxiliary electrodes, while the pseudo-reference electrode was a silver wire. Platinum electrodes were cleaned with the use of flame annealing, and the silver electrode was cleaned with abrasive paper and rinsed with CH_2_Cl_2_. The silver quasi-reference electrode was calibrated using ferrocene, and potentials are quoted relative to the formal potential of the ferrocene/ferrocinium redox couple (E_0′_ = 0.48 V vs. Ag/Ag^+^). The mean of the measured redox potential was taken over five sweeps, and experimental error was estimated to be <20 mV.

### 3.4. UV-Vis Spectroelectrochemical Measurements

UV-Vis spectroelectrochemical measurements were conducted with the use of a HP 8452A spectrophotometer (Agilent Technologies, Santa Clara, CA, USA). A quartz glass cuvette was used as an electrochemical chamber and placed in the path of the light beam. Subsequent UV-Vis spectra were taken for different applied potentials. A glass slide with either indium tin oxide (ITO), platinum coil, or silver wire was used as the working, counter, or reference electrode, respectively. The ITO glass was placed inside the cuvette tightly and separated from the wall using a 100 µm thick Teflon^®^ spacer forming a cell cavity. Special attention was paid to ensure that neither Pt nor Ag electrodes were placed in the optical path. Prior to the measurements, the solutions were bubbled with argon for at least 10 min, to ensure an oxygen-free environment during the collection of data. All measurements were recorded at 25 °C.

### 3.5. EPR Spectroelectrochemical Measurements

In situ EPR spectroelectrochemical experiments on the reduction of H5T in solution were performed in a custom-made glass cylindrical cell, using Pt wire as a working electrode. Measurements were performed using a JEOL JES-FA200 EPR spectrometer (JEOL Ltd., Tokyo, Japan), equipped with a JEOL ES-MCX3B(E) transmission cavity. Prior to the measurements, the solutions were bubbled with argon for at least 10 min, to ensure an oxygen-free environment during the collection of data. All measurements were recorded at 25 °C.

## 4. Conclusions

This work presents complex electrochemical and spectroscopic studies of the model discotic molecule H5T, representing a symmetrically hexa-substituted triphenylene mesogen. The solution studies were focused on deciphering of the redox processes of H5T by combining electrochemical measurements with UV-Vis and EPR spectroscopy. UV-Vis absorption spectroscopy evidences that H5T does not form any aggregates in dichloromethane for concentrations up to 1 × 10^−3^ mol dm^−3^. This is in agreement with the results of cyclic voltammetry, which showed that the first electrochemical process with half-wave potential around 1.05 V represented the reversible radical cation formation. The reversibility of the electrochemical formation of the H5T radical cation was also confirmed by the EPR spectroelectrochemistry. The EPR results evidence that during electrochemical oxidation, stable radicals are formed. A single signal can be observed, which is the result of the unification of signals with very small hyperfine coupling coefficients. The stable paramagnetic intermediates in the first oxidation step are characterized by UV-Vis absorption at around 825 nm. In the second oxidation step, EPR silent dicationic structures are formed.

Moreover, in the frame of the presented study, a tendency of H5T to form molecular assemblies was revealed, with the concentration limit for aggregation in dichloromethane. This was evidenced by the in situ spectroelectrochemical measurements, which showed modified relative absorption maxima as well as a lack of reversibility of the system for a concentration equal to 5 × 10^−3^ mol dm^−3^. This was further supported by the prediction of solvent effects on solute self-assembly in the studied case. In particular, the crucial role of the solvent polarity was revealed, which not only results in better understanding of the solution behavior of anisotropic liquid crystalline molecules, but will also enable aggregation-dependent molecular systems to be designed more efficiently.

## Figures and Tables

**Figure 1 ijms-24-06924-f001:**
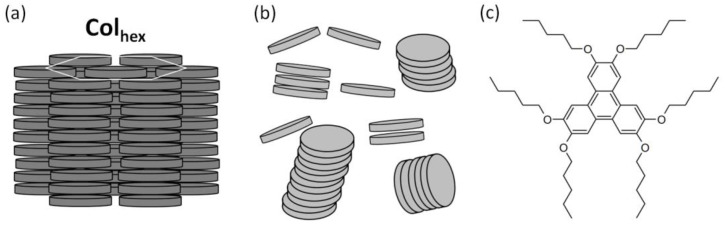
(**a**) Schematic representation of the columnar hexagonal (Col_hex_) mesophase (single-component system). The white hexagon underlines the lateral arrangement of supramolecular columns, with hexagonal symmetry being the most common. (**b**) Schematic representation of the aggregation of discotic molecules in solution (two-component system, under investigation). Solvent molecules not shown. (**c**) Structural formula of 2,3,6,7,10,11-hexapentyloxytriphenylene (H5T), the discotic mesogen under investigation.

**Figure 2 ijms-24-06924-f002:**
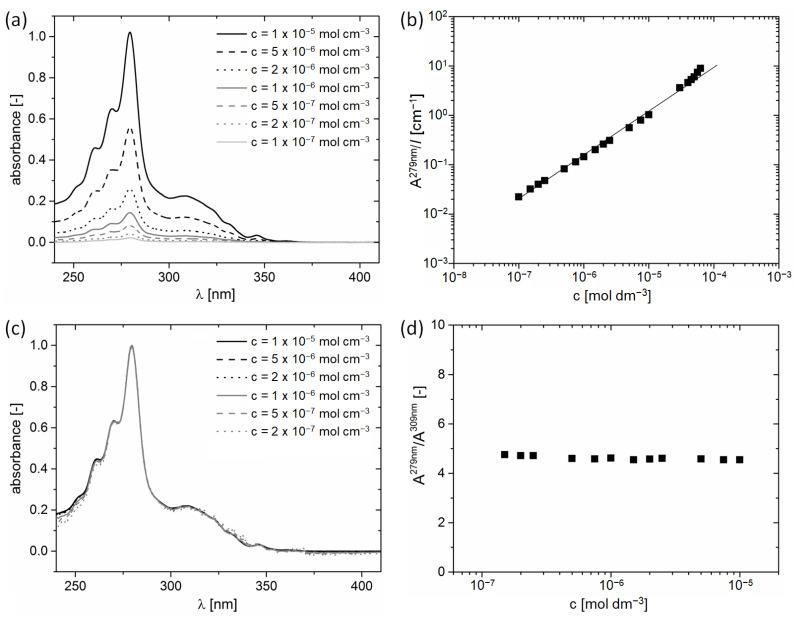
(**a**) UV-Vis absorption spectra of H5T in dichloromethane at 298 K in the range of concentrations 1 × 10^−5^ mol dm^−3^ to 1 × 10^−7^ mol dm^−3^. (**b**) Absorbance at 279 nm/pathlength (*l*) as a function of the concentration of H5T in dichloromethane. (**c**) Normalized UV-Vis absorption spectra and (**d**) ratio of peak intensity of absorption maxima at 279 nm and 309 nm exhibiting a constant value.

**Figure 3 ijms-24-06924-f003:**
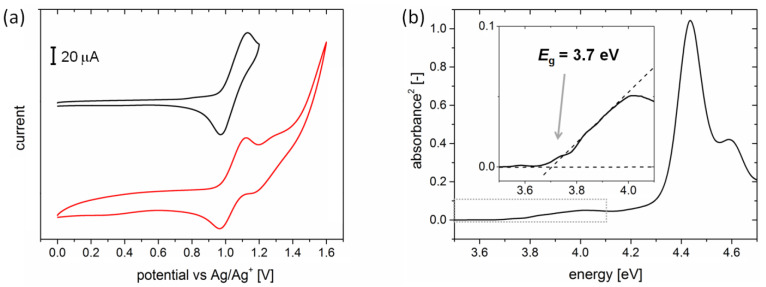
(**a**) Anodic cyclic voltammograms of H5T including first (black curve) and subsequent (red curve) oxidation steps. The sample solution consisted of H5T in dichloromethane (1 × 10^−3^ mol dm^−3^) with 0.2 mol dm^−3^ of *tert*-butylammonium tetrafluoroborate (Bu_4_NBF_4_) as an electrolyte and was scanned at a rate of 0.10 V s^−1^. (**b**) Absorbance squared vs the photon energy used for determination of the gap energy of H5T by extrapolation of the first absorption peak to zero (inset).

**Figure 4 ijms-24-06924-f004:**
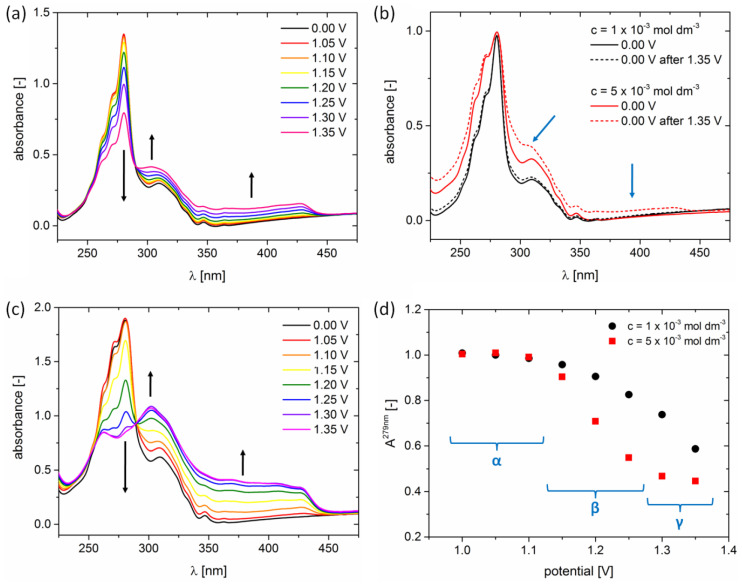
UV-Vis spectroelectrochemistry of (**a**) 1 × 10^−3^ mol dm^−3^ and (**c**) 5 × 10^−3^ mol dm^−3^ H5T solutions in dichloromethane containing 0.2 mol dm^−3^ of Bu_4_NBF_4_ as the supporting electrolyte. The black arrows point out the evolution of spectra. (**b**) Normalized UV-Vis spectra of H5T at 0.00 V, prior to (solid lines) and after (dashed lines) applying 1.35 V. Black and red curves represent 1 × 10^−3^ mol dm^−3^ and 5 × 10^−3^ mol dm^−3^ H5T concentrations, respectively. The blue arrows point out the spectral changes that appeared in the 5 × 10^−3^ mol dm^−3^ solution of H5T after applying 1.35 V. (**d**) Evolution of A^279nm^ absorption maxima of 1 × 10^−3^ mol dm^−3^ and 5 × 10^−3^ mol dm^−3^ H5T concentrations, represented by black dots and red squares, respectively.

**Figure 5 ijms-24-06924-f005:**
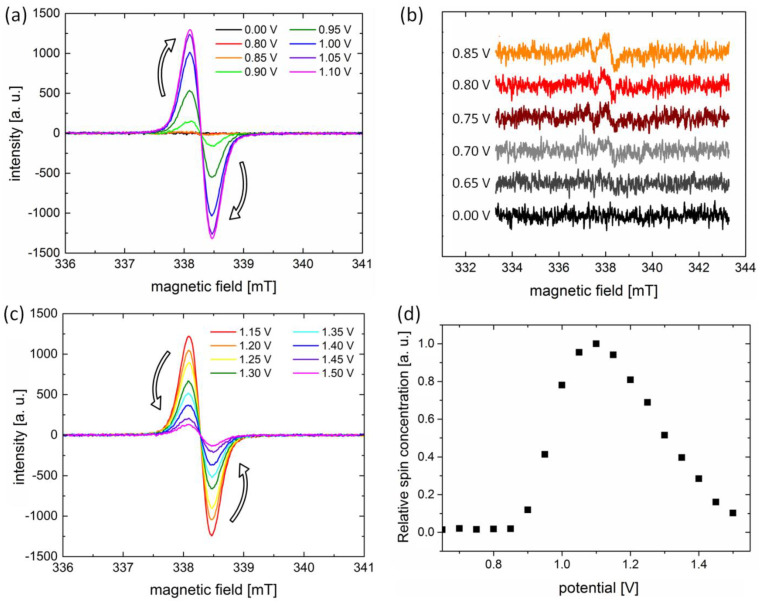
EPR spectroelectrochemistry of the H5T radical cation generated during electrochemical oxidation. (**a**) EPR spectra collected with increasing applied potential. EPR signal appears at 0.90 V and continues to grow for potentials up to 1.10 V. (**b**) Detailed EPR spectra collected in the 0.65–0.85 V range. (**c**) Decrease in EPR signal for 1.15–1.50 V applied potential. (**d**) Changes in the relative concentration of spins generated during oxidation. The sample consisted of 1 × 10^−3^ mol dm^−3^ H5T solution in dichloromethane with 0.2 mol dm^−3^ of Bu_4_NBF_4_ as the supporting electrolyte.

**Figure 6 ijms-24-06924-f006:**
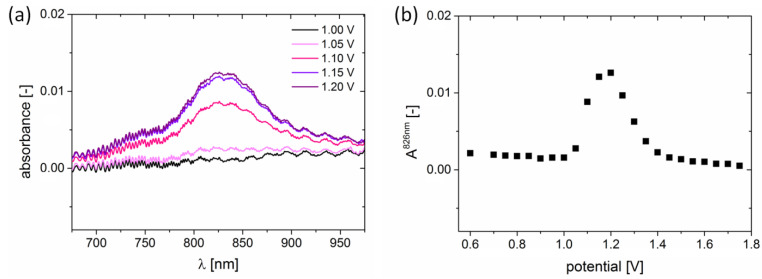
(**a**) UV-Vis spectra evidencing the electrochemical formation of the H5T radical cation. The sample consisted of 1 × 10^−3^ mol dm^−3^ H5T solution in dichloromethane with 0.2 mol dm^−3^ of Bu_4_NBF_4_ as the supporting electrolyte. (**b**) Evolution of A^825nm^ absorption maxima as a function of applied potential.

**Table 1 ijms-24-06924-t001:** Half-wave oxidation potential, HOMO and LUMO energy levels, and optical bandgap of H5T.

*E*_1/2,ox_ (±0.005) (V)	HOMO (eV)	LUMO (eV)	*E*_g_ (eV)
+1.05	−5.37	−1.67	3.70

## Data Availability

No unpublished data were created or analyzed in this article.

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
