# Peer review of "Spectroscopic and Spectroelectrochemical Studies of Hexapentyloxytriphenylene—A Model Discotic Molecule"

_ijms, 2023, doi:10.3390/ijms24086924_

Round 1

Reviewer 1 Report

IJMS-2302928

In this paper, the authors analysed the electrochemical and spectroelectrochemical properties of a pentyloxytriphenylene (HST), were studied by cyclicvoltammetry, UV-Vis and EPR. These measurements were recorded in dichloromethane. The reversible process was evidenced and evaluated the effect of the aggregation.

This paper is interesting, and it is well documented. The introduction is complete but is necessary that the authors reviewer a paragraph. The figures are very correct. Is necessary the Table 1?

Is necessary que in this paper be clearer the new openings of the authors.

Comment

1)      Page 2, Line 64-89, revise the paragraph “Small angle neutron scattering (SANS)…chemical calculations studied [34]”

2)      Comment with major detail EPR spectra

3)      Reference [47]?

4)      Complete the conclusions

Reviewer 2 Report

This paper reports on concentration (and solvent) dependence of aggregation of H5T, a conventional compound as discotic liquid crystal.

Altough importance and limitation of  solvent concentration of aggrigation for H5T (detected by UV-vis spectra, if possible) could be understood clearly, how it is associated with redox behavior (CV) to form radicals (ESR) was not easy to understand. Are the important discovering facts concentration dependence of redox behavor or characteristic behabior of aggrigation compounds?

I am sorry I could not grasp the "main point" of this paper.

Structure of manuscript should be changed to indicate the most important point of this study at the next time.

That's all. 

Reviewer 3 Report

The authors presented a fundamental research article on characterization of a model discotic molecule, 2,3,6,7,10,11-pentyloxytriphenylene (H5T), regarding its spectroscopic and spectroelectrochemical properties at different concentrations. The results implied the critical concentration of H5T for aggregation and reversible radical cation formation, which can potentially provide guidelines for the processability of the molecule in optoelectronic devices fabrication. I would suggest publishing the manuscript if the following comments can be addressed.

  1. Chemical structure of H5T should be provided in Figure 1a with the pi-conjugated polyaromatic core displayed.

  2. A few figures are not described or referenced in the main text, including Figure 1, Figure 4a, Figure 5a and 5c. All figures should be referenced in the main text.

  3. In Line 178, Figure 2S suggested irreversible electrochemical reactions at higher potential. Figure 4b suggested that the irreversible reaction is concentration-dependent. What is the possible irreversible reaction mechanism? Can the authors analyze the structure of the product by NMR or other analytical methods?

  4. What is the difference between the EPR spectra in Figure 5a and Figure 5b? More detailed figure captions should be provided.

  5. The authors discussed solvent effects on self-assembly of H5T but only provided spectroscopic and spectroelectrochemistry data on H5T in one solvent (DCM). UV-vis spectroscopic and/or UV-Vis spectroelectrochemistry data of H5T in solvents of different polarity should be provided.

Round 2

Reviewer 2 Report

After reading additional author's explanation, in this time, I could understand the design of this study (in particular, concentration dependence). Based on reliable experimental facts obtained by established ways in liquid crystal research field, this study and results should be accepted as it is. Sorry.

That's all.

Reviewer 3 Report

The authors have addressed the comments properly and I would recommend publishing the manuscript.